# First observation of the Transient Luminous Events effect on the ionospheric Schumann Resonance, based on China's Seismo-Electromagnetic Satellite

*Shican Qiu[1], Zhe Wang[1], Gaopeng Lu[2], Zeren Zhima[3], Willie Soon[4,5], Victor Manuel Velasco Herrera[6], Peng Ju[1]*

[1]Department of Geophysics, College of the Geology Engineering and Geomatics, Chang'an University, Xi'an, 710054, China

[2]Key Laboratory of Geospace Environment, Chinese Academy of Sciences, University of Science and Technology of China, Hefei, 230026, China

[3]China National Institute of Natural Hazards, Ministry of Emergency Management of P.R.C, Beijing, 100085, China

[4]Center for Environmental Research and Earth Sciences (CERES), Salem, MA 01970, USA

[5]Institute of Earth Physics and Space Science (ELKH EPSS), 9400, Sopron, Hungary

[6]Instituto De Geofísica, Universidad Nacional Autónoma De México, Mexico City, 04510, Mexico

*Correspondence to*: Shican Qiu (*scq@ustc.edu.cn*)

**Abstract.** In this study, we focus on the interactions and interrelated effects among the Transient Luminous Events (TLEs) recorded at the Luoding ground station (22.76°N, 111.57°E), the lightning activities observed by the World Wide Lightning Location Network, and the ionospheric electric field deduced from China's Seismo-Electromagnetic Satellite (CSES). The results show that on September 25, 2021, the signal-to-noise ratio of the Schumann Resonance at the first mode of 6.5 Hz and the second mode of 13 Hz dropped below 2.5 during the TLEs. A significant enhancement of the energy in ELF occurred, and the power spectral density increased substantially. Distinct lightning whistler waves were found in VLF band, further indicating that the energy could possibly be excited by the lightnings. Our results suggest that the observations of electric field from the satellite could possibly be utilized to monitor the lower atmospheric lightnings and their impact on the space environment.

**Keywords**: Transient Luminous Events, lightnings, electric field, Schumann Resonance, whistler wave

## 1 Introduction

The Earth's ground surface to the ionosphere region can be regarded as a spherical resonator, with the dielectric atmosphere in between the electrically conducting layers (Füllekrug and Fraser-Smith, 1996; Schumann, 1952; Surkov et al., 2013). There are diverse electromagnetic activities in the insulated medium, such as lightnings, radioactive particle-induced ionization, ion annihilation, geomagnetic storms, substorms and cosmic ray input, electromagnetic waves generated in wide frequency bands (Füllekrug and Fraser-Smith, 2011; Rycroft et al., 2000; Schumann, 1952). The electromagnetic wave will be reflected when it propagates upward to the highly conductive ionosphere, and finally the wave will be trapped between the surface and the ionosphere (Füllekrug and Fraser-Smith, 2011; Schumann, 1952). Particularly, the lightning activity provides the most important source of electromagnetic field energy (Füllekrug and Fraser-Smith, 2011; Rycroft et al., 2000; Willett et al., 1990). The energy generated from lightnings is balanced with the energy lost during the propagation process, moderated and maintained under a stable intensity (Füllekrug and Fraser-Smith, 2011). In the resonator cavity, some ultra-low frequency (ULF) electromagnetic waves can be consistent with the initial phase after propagating around the Earth for one cycle (Schumann, 1952). This kind of waves with frequencies of 7.8 Hz, 14 Hz, and 20 Hz can resonate with the phase of the initial wave, that is, the Schumann Resonance (SR) (Balser and Wagner, 1962; Sátori et al., 2013; Schumann, 1952). Because of the existence of SR, some bands of the atmospheric electric field energy will show peaks (e.g., the SR frequency) and valleys (e.g., the non-SR frequency) in the spectral graph (Balser and Wagner, 1962; Galejs, 1970; Sátori et al., 2013; Simões et al., 2011). In 1962, the frequencies of SR were deduced to be $\omega_1 = 7.8$ Hz, $\omega_2 = 14.1$ Hz, $\omega_3 = 20.1$ Hz and $\omega_4 = 26.6$ Hz for the first time (Balser and Wagner, 1962). With more and more SR monitoring stations around the world, the global SR distribution has been obtained (Ouyang et al., 2015; Sátori et al., 2013; Zhou et al., 2013). In 2011, observations from the Communication and Navigation Outage Forecast System (CNOFS) satellite showed that the electric field energy of ionospheric F layer increased in some bands, rather consistent with the surface SR mode (Simões et al., 2011). Further, theoretical calculations suggested that the SR can penetrate to the bottom of the ionosphere (Simões et al., 2011; Surkov et al., 2013). Subsequently, the ionospheric SR phenomenon with an energy of $0.5 \mu V/(m * Hz^{1/2})$ was observed by the Chibis-M satellite, and the results proved the existence of the F-layer SR at low and middle latitudes (Dudkin et al., 2014; Simões et al., 2011; Surkov et al., 2013). These results indicated the presence of SR in the ionosphere could also be provided by the energy rooted in the near-surface atmosphere.

On the other hand, the Transient Luminous Events (TLEs) are intense electromagnetic activities occurring in the middle atmosphere, usually generated by tropospheric lightnings located below the TLEs (Boccippio et al., 1995; Franz et al., 1990). The TLEs were determined to be existed and manifested dozens of kilometers above the lightning source regions, requiring strong thunderstorms to energize the electromagnetic waves (Boccippio et al., 1995; Franz et al., 1990; Sentman et al., 1995). According to their shapes and colors, TLEs are usually classified into six types as sprites (Mende et al., 1995; Sentman et al., 1995), elves (Fukunishi et al., 1996), blue jets (Wescott et al., 1995), blue starters (Wescott et al., 1996), gigantic jets (Pasko et al., 2002) and halos (Stenbaek-Nielsen et al., 2000). The mechanism of sprites and elves involves the electromagnetic field

heating the particles at the bottom of the ionosphere, which will thus provide a feedback of huge amount of energy through differential potentials (Boccippio et al., 1995; Mende et al., 1995; Sentman et al., 1995). The TLEs are indeed found to be closely related to atmospheric and ionospheric electromagnetic activities (Bösinger et al., 2006; Sátori et al., 2013; Shalimov et al., 2011). For example, the spectral structure of the surface SR has been studied by the sprite Q-burst, the electrical impulses generated during TLEs have been analyzed, and the ionospheric Alfvén wave resonances excited by jets have been monitored

(Bösinger et al., 2006; Füllekrug et al., 1998; Guha et al., 2017). The Alfvén waves are considered to be controlled by global lightning activity, which will affect the magnetic field in the ionospheric region (Bösinger et al., 2002; Surkov et al., 2013).

In addition, whistler waves could also be generated during lightning events and transmitted to the ionosphere (Bayupati et al., 2012; Carpenter and Anderson, 1992; Holzworth et al., 1999; Storey, 1953). In 1990s, with the launch of electromagnetic satellites, lightning whistlers were observed in the ionosphere, confirming that the huge energy can be coupled to the F layer

and even penetrate to the ionosphere (Bayupati et al., 2012; Carpenter and Anderson, 1992; Holzworth et al., 1999). The whistler wave generally can be recognized by a falling frequency (Bernard, 1973; Carpenter and Anderson, 1992; Helliwell, 1965; Helliwell and Pytte, 1966). Intelligent algorithms can now be applied to filter whistler waves from satellite data and locate the lightning source regions (Dharma et al., 2014; Lichtenberger et al., 2008).

Therefore, TLEs can be used as an important carrier or tracer to study the structural coupling of atmospheric layers.

However, there are still many gaps in research on TLEs because it is relatively difficult to observe them. On the other hand, the ionospheric SR energy comes from the surface and is closely related to the space environment. In this study, we utilize the latest data of ionospheric electric field from China's Seismo-Electromagnetic Satellite (CSES), through the methods of the signal-to-noise ratio (SNR) and the power spectral density (PSD) for the first time, to study the relationship between TLEs and ionospheric SR.

## 2 Materials and Methods

The TLEs captured/observed from the Luoding monitoring station will be analyzed. The electromagnetic field data come from the CSES satellite, orbiting at an altitude of 500 km around the F layer of the ionosphere (Diego et al., 2020). It is designed to adopt a 5-day revisit cycle, with a daily progression of 500 km eastward (Diego et al., 2020). The data length of each sampling point in the ELF band is 256, and the sampling frequency is about 100 Hz with an interval of 8 ms (Diego et

al., 2020). Each sample point of VLF band has a length of 2048, a frequency of 12 kHz, and an interval of 80 μs (Diego et al., 2020). The lightning observations are provided by the World Wide Lightning Location Network (WWLLN), recording the time, latitude and longitude of the lightnings (Jacobson et al., 2006). In order to obtain the corresponding ionospheric SR characteristics, the electric field from the CSES satellite data with the closest orbit is utilized.

The methods for data processing are the signal-to-noise ratio (SNR) and the power spectral density (PSD). The SNR is a

95 method proposed to study the field perturbations (Molchanov et al., 2006). Its main principle is to measure the amplitude of

the perturbation through energies from a specific band relative to the chosen or reference rest bands. The calculation of SNR is given as

$$SNR = \frac{2A(f_0)}{A(f_-) + A(f_+)} \tag{1}$$

where $A(f_0)$ is the energy corresponding to the signal $f_0$, $A(f_-)$ is the noise energy whose band is lower than the SR signal, and $A(f_+)$ is the noise energy with band higher than the SR signal. Then we can set the peak frequency of SR as the signal $f_0$, the field frequency lower than SR as $f_-$, and larger than SR as $f_+$.

The principle of the PSD method is spectrum analysis from the Fourier transform (Bracewell and Kahn, 1966; Bracewell, 1989). The frequency domain information of each sampling point is calculated to obtain the spectrum. Then the frequency domain data are arranged in the order of time or space to draw a three-dimensional PSD map. The calculation of Fourier transform is given as:

$$F(s) = \int_{-\infty}^{+\infty} f(x)e^{-i2\pi xs}dx \tag{2}$$

where $F(s)$ represents the frequency domain and $f(x)$ is the time domain.

## 3 Observations and Discussion

The TLEs captured by the Luoding station in 2021 is shown as the orange star in Figure 2. The TLEs were mainly concentrated from May to September, consistent with the local thunderstorm season. We focus on the case on September 25, 2021 to study the ionospheric SR anomalies during the TLEs. A total of 36 TLEs were photographed between 16:00 and 20:00 UT. Some of the more prominent TLEs are shown in Figure 1a to 1f, taken at 17:59, 18:20, 18:38, 18:53, 19:01, 19:49, and 20:07 UT, respectively. All of the represented TLEs are classified as the red sprites.

On the other hand, the observations from WWLLN exhibited huge thunderstorm activities developing near the Hainan Island. The lightnings were mainly concentrated in the range of 18-22°N, 110-113°E, 200 km southeast of Luoding station. Around 15:00 UT, the lightnings appeared and maintained until 22:00 UT. Therefore, the duration of the thunderstorm included the main occurrence time of the TLEs over Luoding. The pink dots in Figure 2b and 2e exhibited the lightnings mainly manifested within the 17:00 to 20:00 UT time window.

Comparing the CSES orbit time with the occurrence of TLEs, we selected the nighttime side data of NO. 20239 orbit for analysis. The CSES's orbital position passed near the Luoding station at around 18:33 UT, located on the west side with a horizontal distance of about 200 km. From September to October, around the same orbital positions, i.e., the repeat of the satellite's orbit around Luoding after every five-day cycle, were selected for the background data. The appropriate CSES satellite orbits were found on September 20 and October 15. No TLEs were captured on October 15. And on September 20, TLEs were only photographed during the 13-15 UT period, with no TLEs appeared when CSES traveled overhead Luoding

station at 18:00 UT. The lightnings on September 20 and October 15 are also shown in Figure 2a and 2c. It can be seen and confirmed that no large-scale thunderstorm area on September 20 and October 15.

Taking the SR as the signal and the non-SR as the noise, the variations of SNR with latitude can be calculated. Through evaluation and analysis, the first mode of SR is selected to be 6.5 Hz, while 3 Hz and 8 Hz are adopted accordingly as the upper and lower bands. The second mode of SR is selected to be 13 Hz, with 10 Hz and 16 Hz as the upper and lower noise bands, respectively. According to the SNR calculation method from Eq. 1, the background SNR is shown is Figure 2a, 2d, 2c and 2f. In comparison, the SNR for the CSES's orbit NO. 20239 is given by Figure 2b and 2e. The red, green and blue circles indicate SNR > 5, 2.5 < SNR < 5, and SNR< 2.5, respectively. Figure 2a, 2d, 2c, and 2f demonstrate that the background SR has SNR above 2.5, with many of the sampling points even above 5, when there is no disturbance. Figure 2b and 2e show the SR's first and second modes for the NO. 20239 orbit, respectively. The decrease of SNR is mainly concentrated in the area between 10 ~ 22°N, shown as the blue circles. This region is close to the lightning area developing near the Hainan Island, with the point at 18°N located just over the thunderstorms. Accompanied by the occurrence of a large number of lightnings and TLEs, the SNR is significantly reduced to below 2.5, with the first mode of 1.9 and second mode of 1.2, respectively.

Figure 3 shows the energy distribution of all bands in the range of 1-20 Hz on the three days, exhibiting the energy variation with frequency and latitude, i.e., the PSD. It includes the first mode of the SR at 6.5 Hz, and the second mode at 13 Hz. Figure 3a and 3c represent the PSDs on September 20 and October 15, respectively, when both the lightning activties and TLEs were weak. In Figure 3a and 3c, the electric field energy is concentrated in the two bands of SR, which is much higher than that of the non-SR bands. The peak frequency of SR maintains stable, and the overall variation does not exceed 0.5 Hz. However, when the TLEs appear, the PSD in Figure 3b exhibits obvious perturbations. The electric field energy recorded by CSES's orbit NO. 20239 is not concentrated in the SR band, but distributed and spread more broadly in the band from 1 to 20 Hz. Within the 10-22°N region on September 25, the PSD is significantly enhanced, especially in the non-SR bands. In the 23-30°N region, the PSD still maintains two distinct SR modes, which is consistent with the background as shown in Figure 3a and 3c. The peak field strengths of the first and second SR modes are about $0.5\mu V/(m*Hz^{1/2})$, which is consistent with the Chibi-M satellite data (Dudkin et al., 2014; Simões et al., 2011). It can be found that the enhanced region in Figure 3b is consistent with the region of decreasing SNR in Figure 2b and 2e. Therefore, we believe that during the NO. 20239 orbit, the ionospheric F layer in the 10-22°N region has undergone significant disturbances due to the combined action of both lightning and TLE activities.

We can further verify that the low-Earth orbiting satellite can record the electromagnetic energy from lightnings and TLEs coupled to the top of the ionosphere. Figure 4 shows the lightning whistler waves recorded by the electric field meter (a) and the magnetometer (b) during the NO. 20239 orbit around 18:31 UT, when the CSES satellite was located at 16.5°N. The whistling waves recorded by CSES range from 16-20°N, which coincides with the distribution area of the lightning activities. The whistling wave durations are within 1-2 s, and the frequencies are generally in the range of 3-10 kHz, which is consistent with a typical whistling wave phenomenon (Bayupati et al., 2012; Carpenter and Anderson, 1992; Holzworth et al., 1999). As the high-frequency whistler wave has a higher group velocity and will be the first to be received, it shows the characteristic

falling frequency in the PSD. These whistler wave signals indicate that the electromagnetic field generated by lightnings and TLEs can be superimposed on the ionospheric electric field, resulting in an increase in the energy of the electric field in a certain band.

Furthermore, we analyze the latitude-dependent energy in each band of the ELF, as shown in Figure 5. Among them, the 18.07 Hz is the third mode of SR with weaker energy (shown as the red line), 12.69 Hz is the second mode of SR (blue), and
165 6.34 Hz is the first mode (green). The solid line represents the measured electric field during CSES's orbit NO. 20239, the dashed line exhibits the observation on September 20, and the dotted line denotes the electric field on October 15. It can be found that between the 10-22°N region, the energy in each band of the NO. 20239 orbit is much stronger, while the energy in the three days is almost similar between the region of 23-30°N. It is calculated that the increase of the energy in each band within the 10-22°N region is 300% at 6.34 Hz, 270% at 12.69 Hz, and 130% at 18.07 Hz. The energy of the first mode (i.e.,
6.34 Hz) and the second mode (i.e., 12.69 Hz) of the SR are enhanced significantly, while the energy of the faint third mode also increases obviously. Therefore, the decrease of SNR during the TLEs should reasonably be caused by the energy increase in the non-SR band.

The electric field energy of the three bands has a maximum enhancement near 17.5°N, very close to the range of lightning activity in Figure 3b. At the same time, the electromagnetic field of the VLF band in this area also records the presence of
175 whistler waves (Figure 4). Therefore, we believe that the electric field disturbance of the 20239 orbit is caused by lightning and TLEs activity. The results show that when lightning and TLEs occur, the electric field with lower frequency is more likely to penetrate to the high conductance region of the ionosphere. These energies cause the ionospheric electric field to be disturbed, the energy of each band of ELF increases, and the SNR of SR decreases. That is, the electric field generated during lightning and TLEs superimposes energy on each band of the ionosphere, just as whistler wave energy superimposes on the VLF band.
In order to make the new results more credible, we further examine and study the variation of the geomagnetic Dst index for September 2021, and the large-scale geological activities. It is found that there was no strong geomagnetic storm around September 2021, nor obvious any prominent geological activity near Luoding. In order to exclude the influence of other factors, we examined the possible influence of solar activity and high-energy proton events. The data showed that the solar activity was relatively calm during these days, and there were also no high-energy particle events. Therefore, based on the observations
of lightnings and TLEs, we conclude that the ionospheric anomalies on September 25, 2021 could possibly be controlled by these two factors. The TLEs captured were all red sprites, whose formation mechanism is the electromagnetic pulse transferring energy to the bottom of the ionosphere. Then the TLEs may cause dramatic changes in the ionospheric particles and obvious disturbances in electromagnetic field energy. However, the lightning activities will also produce Alfvén waves, which are close to the two modes of SR, and may interfere with the ionospheric SR (Beggan et al., 2018; Dudkin et al., 2014). Therefore, we
further analyzed the time-frequency characteristics of the interference patterns between Alfvén wave and the Schumann resonance, comparing with the electric field disturbance. The interference pattern is generally manifested as specific fingerprint shape. However, the SR anomaly observed in Fig.3 is quite different from this particular interference characteristics. Thus, we believe that this anomaly is not caused by Alfvén wave.

The lightning whistler wave suggests that the electromagnetic energy will be transported to the altitude range of the satellite orbit. The specific coupling process could possibly be proposed as follows: the lightnings and TLEs generate electromagnetic waves in multiple bands; in the process of upward coupling, electric fields in many bands can transmit energy to the ionosphere; when these electromagnetic waves penetrate to the ionosphere, energies can be coupled to the satellite orbit around the F layer of the ionosphere. Since the electromagnetic field has been generated in both SR band and non-SR band, the energy increase of multiple bands is observed by the CSES satellite. In the ionospheric background, the SR energy is supplied by the lower boundary near the surface region, manifested as the peak energy in specific bands (Figure 3a and 3c). When the perturbation of electromagnetic field from the lightning activities and TLEs is superimposed on the ionosphere, the energy of each band will increase obviously in the PSD (Figure 3b and Figure 5). The energy proportion of SR will decrease, leading to the decrease of the SNR. Since the energy attenuation in the low-frequency electric field is small, for the ELF band the stronger energy of the electric field is, the stronger the SR perturbation will be. In the VLF band, the electric field meter and magnetometer recorded the lightning whistler waves; while in the ELF band, the electric field energy shows an anomalous increase. Besides, the remaining 13 cases of TLEs documented near the Luoding station were also analyzed, and 6 cases of them appeared near Luoding station close to 18:30. In 2 cases, SR data were recorded by CSES, which were exhibited similar perturbations. Therefore, this perturbation may be widespread during TLEs activity.

**4 Conclusions**

In this research, the latest ionospheric electric field data from the CSES satellite are utilized for the first time to the study the disturbance of ionospheric SR during lightning and TLE events. The results show that the disturbances of the ionospheric electric field can be coupled upward to the CSES's orbital altitude, and lead to strong SR perturbations. The electric field variations can be represented and studied in terms of PSD and SNR diagnostic metrics. The disturbance of SR is mainly in the ELF band, while the whistling wave is in the VLF band. Our results suggest the existence of a coupling to the ionospheric F-layer via lightnings and TLEs.

The conclusions are given as follows:

1. A huge lightning and TLEs event co-occurred on September 25, 2021, penetrated to the bottom of the ionosphere and caused electric field disturbance in the upper satellite orbit region. During the large lightning activity, the horizontal range of this disturbance can reach hundreds of kilometers.

2. The lightning activities and TLEs will increase the PSD energy of the ionospheric electric field and reduce the signal-to-noise ratio of the first and second modes of Schumann resonance. Comparing the Alfvén wave with the SR interference phenomenon, we believe that it is likely that this disturbance is indeed caused by the electric field originated from lightnings, instead of the Alfvén wave.

3. When the SR disturbance occurs in the ELF band of the electric field, the whistler wave signal is detected in the VLF band. The whistler wave indicates that the electric field energy generated by the lightning and TLEs can cause the disturbance to altitudes as high as the CSES's satellite orbit.

**Data availability**

The dataset of Transient luminous events over high-impact thunderstorm systems comes from the National Space Science Data Center, DOI:10.12176/01.05.00070-V01, https://vsso.nssdc.ac.cn/nssdc_en/html/vssoinfo.html?1374, accessed on 28 October 2023. The Coordinated Observations of Transient Luminous Events is available from the National Space Science Data Center, DOI:10.12176/01.05.00069-V01, https://vsso.nssdc.ac.cn/nssdc_en/html/vssoinfo.html?1370, accessed on 28 October 2023. The ionosphere data of China's Seismo-Electromagnetic Satellite (CSES) can be downloaded from its database (https://leos.ac.cn, accessed on 28 October 2023), via registering to select the specific kind of data. The lightning location and power data are available from the World Wide Lightning Location Network (http://wwlln.net/, accessed on 28 October 2023), via internet with cadence every 10 minutes for research purposes from the University of Washington.

**Acknowledgements**

This work is supported by the Fundamental Research Funds for the Central Universities, CHD (NO. 300102263205), and the West Light Cross-Disciplinary Innovation team of Chinese Academy of Sciences (NO. E1294301). This work has made use of the data from China's Seismo-Electromagnetic Satellite (CSES) mission, a project funded by China National Space Administration (CNSA) and China Earthquake Administration (CEA). We acknowledge for the data resources from the National Space Science Data Center, National Science and Technology Infrastructure of China. We thank the World Wide Lightning Location Network (WWLLN).

**Author information**

**Affiliations**

**Department of Geophysics, College of the Geology Engineering and Geomatics, Chang'an University, Xi'an, 710054, China**

Shican Qiu, Zhe Wang and Peng Ju

**Key Laboratory of Geospace Environment, Chinese Academy of Sciences, University of Science and Technology of China, Hefei, 230026, China**

Gaopeng Lu

**China National Institute of Natural Hazards, Ministry of Emergency Management of P.R.C, Beijing, 100085, China**

Zeren Zhima

**Center for Environmental Research and Earth Sciences (CERES), Salem, MA 01970, USA**

Willie Soon

**Institute of Earth Physics and Space Science (ELKH EPSS), 9400, Sopron, Hungary**

Willie Soon

**Instituto De Geofísica, Universidad Nacional Autónoma De México, Mexico City, 04510, Mexico**

Victor Manuel Velasco Herrera

## Contributions

Shican Qiu conceived this study and wrote this manuscript.

Zhe Wang performed data analysis and prepared the figures.

Gaopeng Lu added some materials about Transient Luminous Events and lightning strokes.

Zeren Zhima supplied some data of China Seismo-Electromagnetic Satellite

Willie Soon was in charge of the organization and English polishing of the whole manuscript.

Victor Manuel Velasco Herrera added to data processing and analyses.

Peng Ju added some materials in the discussion.

## Competing interests

The authors declare no conflict of interest.

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

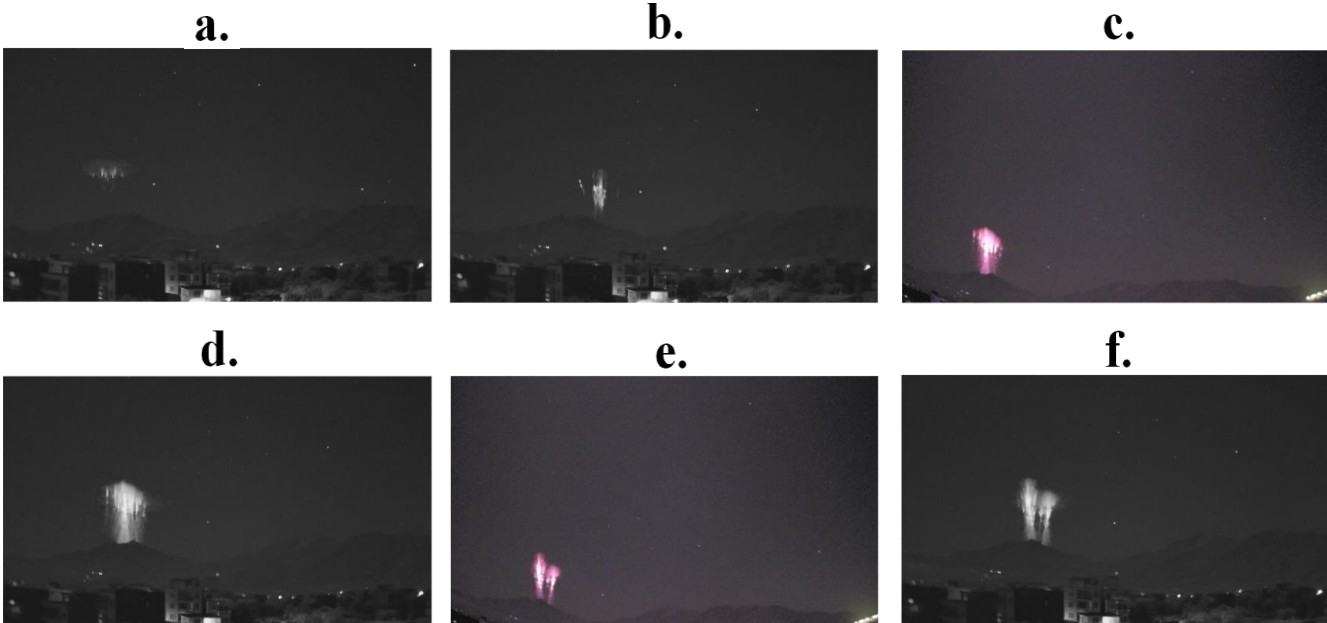

**Figure 1: The transient luminous events (TLEs) captured around the Luoding station on September 25, 2021, at a. 17:59, b. 18:20, c. 18:38, d. 18:53, e. 19:01, f. 19:49, and g. 20:07 UT, respectively.**

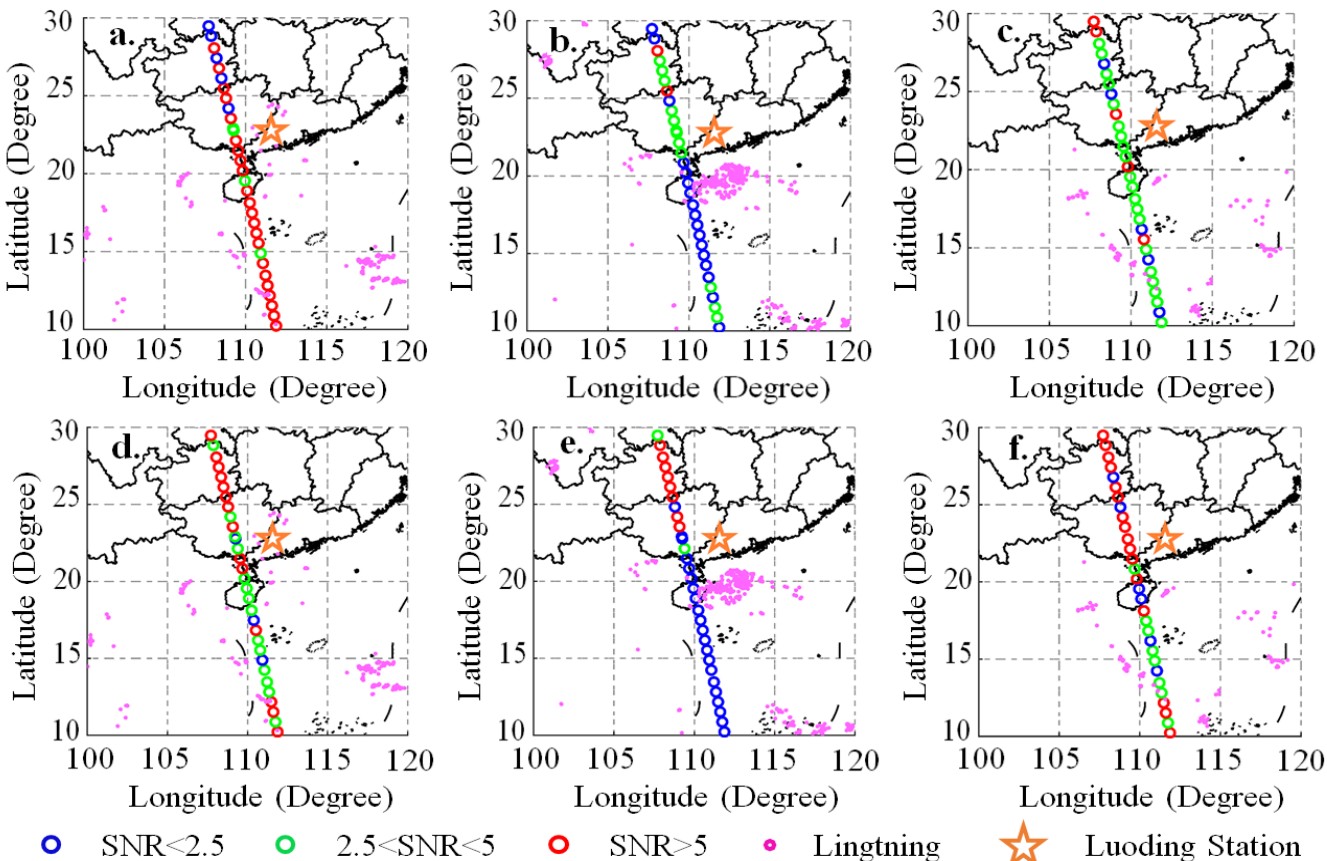

Figure 2. The signal-to-noise ratio (SNR) for the Schumann resonance (SR) of the ionospheric electric field and the distributions of the lightnings. The red, green and blue circles indicate the SNR > 5, 2.5 < SNR < 5, and SNR < 2.5, respectively. The pink dots point out the lightning locations. The orange pentagram represents the Luoding station. a. The SNR of the first mode on September 20, 2021, and the lightnings from 17:00 to 20:00 UT. b. The SNR in the first mode of NO. 20239 orbit of CSES on September 25, 2021, and lightning distributions. c. The SNR of the first mode on October 15, 2021, and lightning distributions. d. The SNR of the second mode on September 20, 2021. e. The SNR of the second mode of NO. 20239 orbit. f. The SNR of the second mode on October 15, 2021.

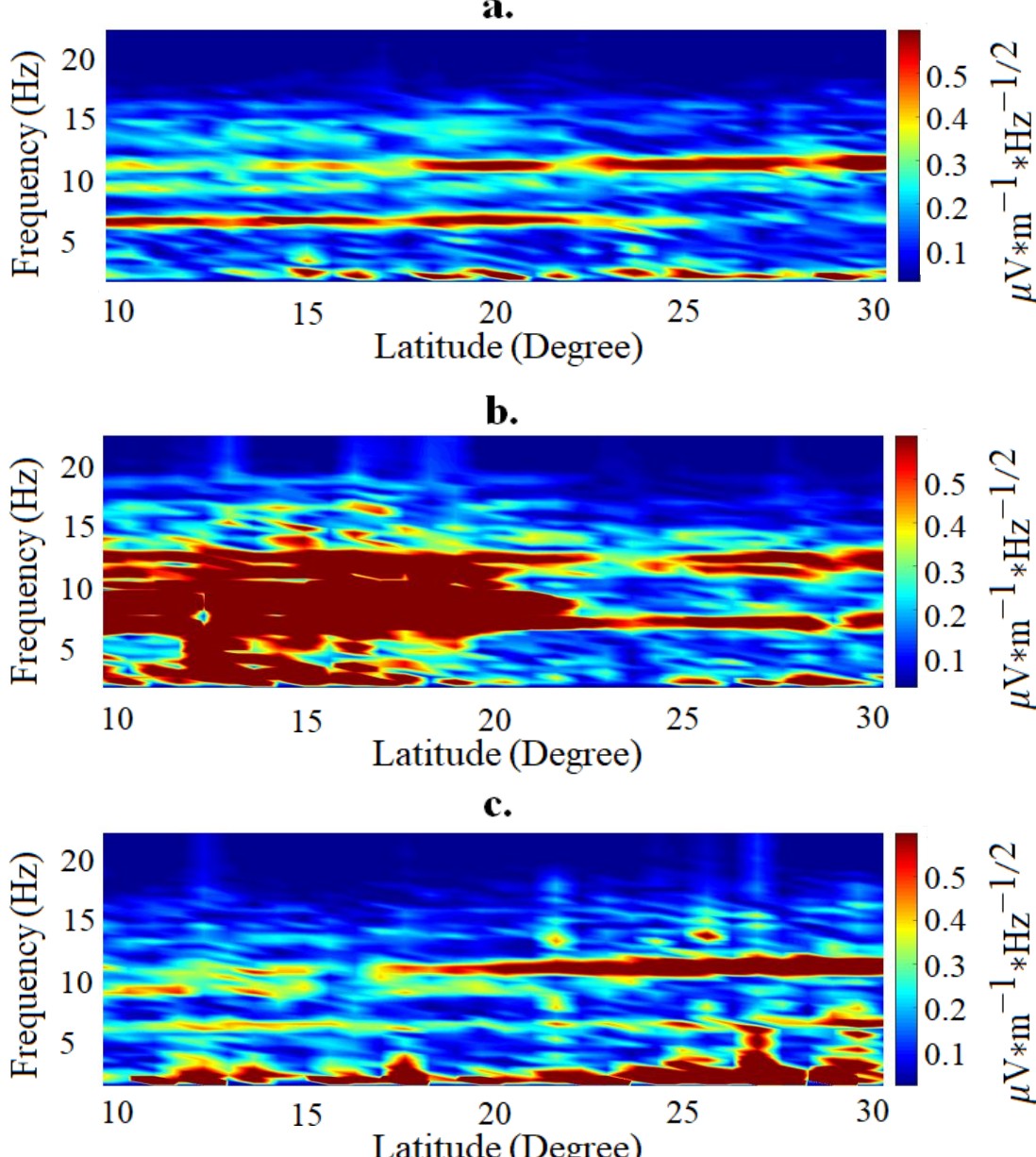

**Figure 3. The Power Spectral density (PSD) of the ionospheric electric field SR. a. The electric field PSD on September 20, 2021. b. The electric field PSD of the CSES's orbit NO. 20239 on September 25. c. The electric field PSD on October 15. In a. and c., the PSD has two obvious peak lines at 6.5Hz and 13Hz as the first and second modes of the SR, respectively.**

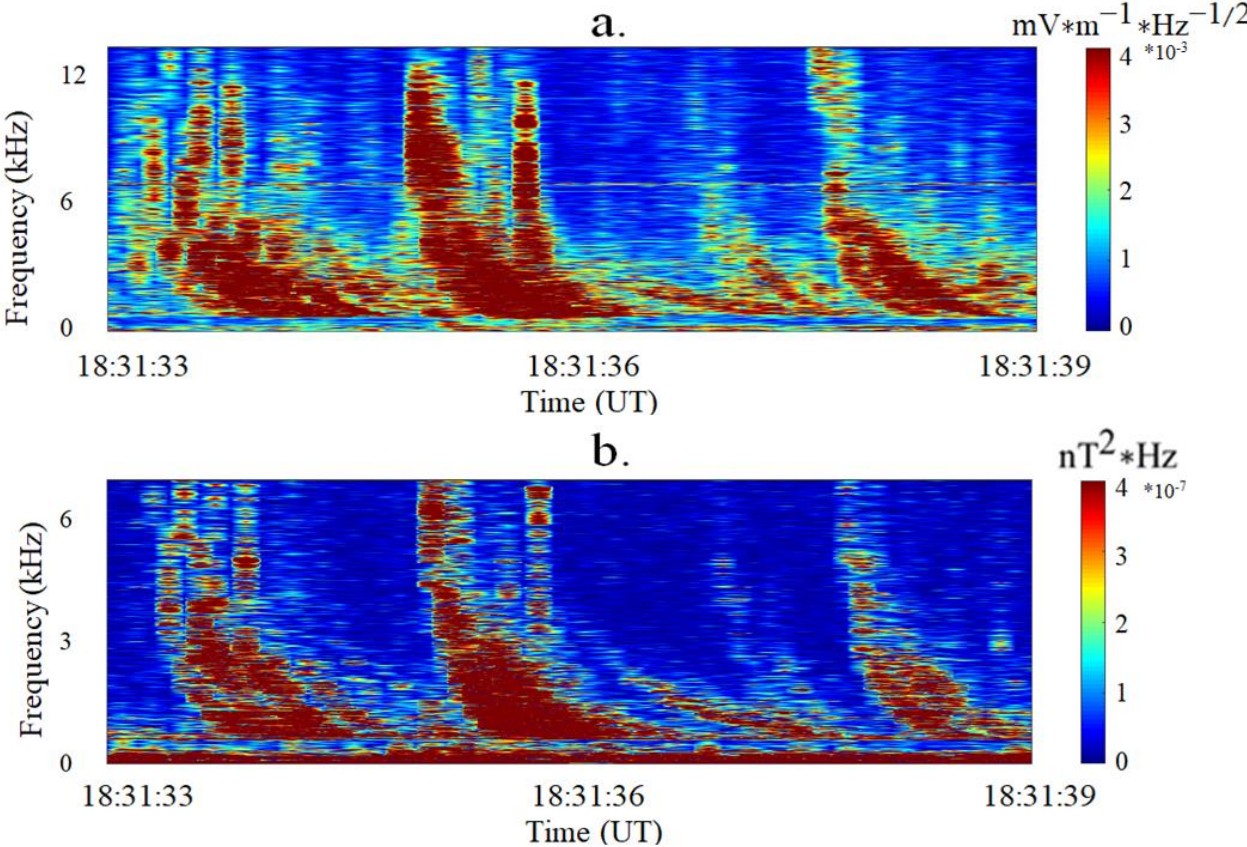

**Figure 4. The lightning whistler waves in obvious falling frequency displayed during orbit NO. 20239, observed on 18:31:36 UT at 16.6°N. a. The lightning whistler waves from the electric field meter. b. The lightning generated whistler waves power spectral density observed by the magnetometer.**

400

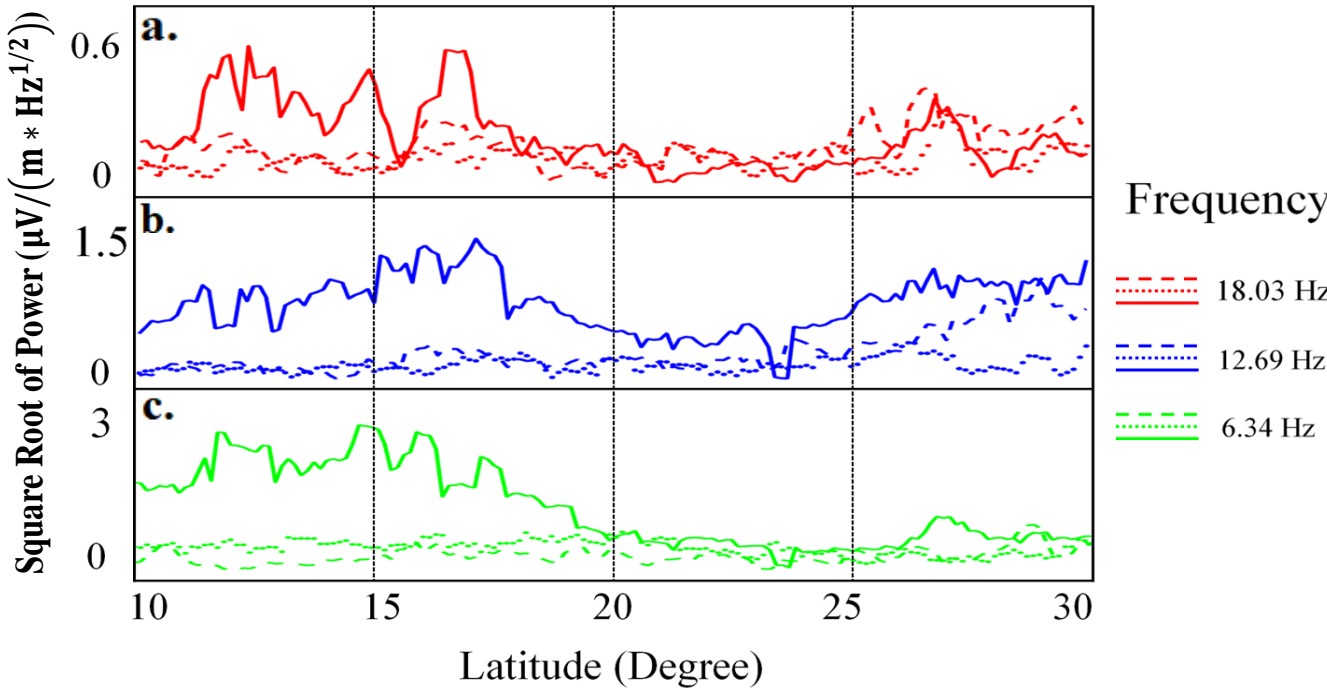

**Figure 5.** The energy variations of the electric field in each ELF band, from CSES's orbit NO. 20239. The lines in red, blue and green colors represent the energy intensity at 18.07 Hz, 12.69 Hz and 6.34Hz, respectively. The solid line is the measured electric field during orbit NO. 20239 on September 25. The dashed line indicates the electric field on September 20, and the dotted line denotes the electric field on October 15.