# Peer review of "First observation of the Transient Luminous Events effect on the ionospheric Schumann Resonance, based on China's Seismo-Electromagnetic Satellite"

_EGUsphere, 2024_

## Author Comment (AC1)

We would like to express our cordial thanks to the reviewer for the valuable comments and constructive suggestions. We appreciate the input of the reviewer and hope to make sure that we are incorporating his or her significant inputs to strengthen our analysis. We went through all the comments thoroughly and revised the manuscript accordingly.

This paper reports what could be a chance coincidence, at ~ 18.30 UT on 25 September 2021. This is a relation between sprites (which are known to perturb the ionospheric D-region above them - references to that should be given) and claimed Schumann resonance (SR) signal changes observed on the CSES satellite. This is in a Sun-synchronous orbit at ~ 500 km altitude. The strength of SR signals at such a height needs to be estimated using magnetoionic theory. The SR changes shown are dramatic, from the first two harmonic signals, at 8 and 14 Hz, to a broadband signal. How this might happen needs justification.

1. The units of the signal strengths shown in Fig. 3 and 4 are incorrect. If squared, the units in Fig. 4b would be correct, I think.

Reply: Thanks for your valuable comment. The units we used for the electric and magnetic field signal strengths in Figure 3 and 4 are those originally provided by the China Seismo-Electromagnetic Satellite (CSES) database. We compared the previous references, (such as 'Satellite observations of Schumann resonances in the Earth's ionosphere' (Simões et al., 2011), and 'Electric field signatures of the IAR and Schumann resonance in the upper ionosphere detected by Chibis-M microsatellite' (Dudkin et al., 2014)), and found they also use the Schumann resonance signal unit of $\mu$V*m$^{-1}$*Hz$^{-1/2}$. The Schumann resonance strengths in the data from the CSES are consistent with the Schumann resonance strengths in these references. Therefore, we have to keep the data unit provided by CSES to characterize the intensity of Schumann resonance. The electric field in ULF, electric field in VLF and magnetic field in VLF provided by CSES can be accessed through the official website of https://leos.ac.cn/#/dataService/dataDownloadList.

2. The fractional hop whistlers shown in Fig. 4 do not have a characteristic "L" shape - see the monograph by Helliwell (1965). What is the duration of the record shown in Fig. 4? How many seconds?

Reply: Thanks for your valuable comment. Due to the large sampling intervals, the resolution of the graph is low. However, if we shorten the time of the plot, we can see very wide 'L'-shaped whistling waves as shown in Figure 1* below. It can be found that a 2s-whistling wave occurring at 18:31:36 UT, and a 1s-whistling wave occurring at 18:31:38 UT. The durations of the whistle waves in the Figure 1* are about 1-2 s, which is consistent with the typical characteristics of single-hop and two-hop whistle waves. Since the result is consistent with the duration of whistling waves studied by previous researchers, we believe that there is a lightning whistling wave phenomenon generated near the Hainan Island. We will redraw the time-frequency diagrams of the whistling waves in the main text.

[Figure]

Figure 1*. The lightning whistling wave in obvious "L" form displayed during the NO. 20239 orbit, observed on 18:31:36 UT at 16.6°N.

3. What is the bandwidth of the filters used to generate Fig. 5. What determined the central frequencies used? Why are these values shown to two decimal places? These are ELF signals, not ULF.

Reply: Thanks for your valuable comment. We use the filter bandwidth provided by the satellite data with sampling frequency of 0.5 Hz. The central frequency of the electric field signal can be also obtained through calculation based on the satellite sampling time. The ULF band frequencies in the paper correspond to the frequencies of the electric field level 2 data provided by the CSES, which can be accessed through the CSES data website (https://leos.ac.cn/#/dataService/dataDownloadList).

The CSES database defines the signal of 0-20 Hz as ULF, the signal of 100-2000 Hz as ELF, and the signal of 2k-24k Hz as VLF. As a results, in this paper, such frequency classification is used to avoid confusions caused by different definitions of the ULF band by other standards and researchers.

4. In Section 3, much of the discussion is too general and vague.

Reply: Thanks for your valuable comment. We have modified section 3 carefully and added some discussions of the results.

We have replaced lines 116-128 with this revised and more detailed texts:

'Taking the SR as the signal and the non-SR as the noise, the variations of SNR with latitude can be calculated. Through evaluation and analysis, the first mode of SR is selected to be 6.5 Hz, while accordingly 3 Hz and 8 Hz are adopted as the upper and lower bands. The second mode of SR is selected to be 13 Hz, and 10 Hz and 16 Hz as the upper and lower noise bands, respectively. According to the SNR calculation method from Eq. 1, the background SNR is shown in Figure 2a, 2d, 2c and 2f. In comparison, the SNR for orbit NO. 20239 is given by Figure 2b and 2e. The red, green and blue circles indicate SNR > 5, 2.5 < SNR < 5, and SNR< 2.5, respectively. Figure 2a, 2d, 2c, and 2f demonstrate that the background SR has SNR above 2.5, with many of the sampling points even above 5, when there is no disturbance. Figure 2b and 2e show the SR's first and second modes for the NO. 20239 orbit, respectively. The decrease of SNR is mainly concentrated in the area between 10 ~ 22°N, shown as the

blue circles. This region is close to the lightning area near the Hainan Island, with the point at 18°N located just over the thunderstorms. Accompanied by the occurrence of a large number of lightnings and TLEs, the SNR is significantly reduced to below 2.5, with the first mode of 1.9 and second mode of 1.2, respectively. It can be found that in the presence of lightnings and TLEs, the SNR of the selected SR is generally reduced below 2.5. This result indicates under the influence of lightnings and TLEs, the SR signal of the ionospheric electric field will become unstable, and its SNR could reduce by half.'

In order to describe the whistling wave we observed, we have added statements around line 144 as: 'The whistling waves recorded by CSES range from 16-20°N, which coincides with the distribution area of the lightning activities. The whistling wave durations are within 1-2 s, and the frequencies are generally in the range of 3-10 kHz, which is in consistent with a typical whistling wave phenomenon (Bayupati et al., 2012; Carpenter and Anderson, 1992; Holzworth et al., 1999).'

Since the frequency of the Alfvén wave is close to the ULF band, the Alfvén wave generated by the lightning may also cause a perturbation of the electric field. In order to eliminate the Alfvén wave, we have added statement around line 170 as: 'The lightnings will also produce Alfvén waves, which are close to the two modes of SR, and may interfere with the ionospheric SR (Beggan et al., 2018; Dudkin et al., 2014). Therefore, we analyzed the time-frequency characteristics of the interference patterns between Alfvén wave and the Schumann resonance, comparing with the electric field disturbance. The interference is generally manifested as a fingerprint shape. However, the SR anomaly in Fig.3 is quite different from this interference characteristics. Thus, we still believe that this anomaly is not caused by Alfvén wave.'

Furthermore, in order to make the background data more perfect, we excluded the influence of solar flares and cosmic rays, we state around line 166: 'In order to exclude the influence of other factors, we examined the possible influence of solar activity and high-energy proton events. The data showed that the solar activity was relatively calm during these days, and there were no high-energy particle events.'

We hope the modified Section 3 is sufficiently or adequately improved.

5. The three main conclusions need to be redrafted.

Reply: Thanks for your valuable comment. We revised the conclusion as follows:

1. A huge lightning and TLEs event co-occurred on September 25, 2021, penetrated the bottom of the ionosphere and caused electric field disturbance in the upper satellite orbit region. During the large lightning activity, the horizontal range of this disturbance can reach hundreds of kilometers.

2. The lightning and TLEs will increase the PSD energy of the ionospheric electric field and reduce the signal-to-noise ratio of the first and second modes of Schumann resonance. Comparing the Alfvén wave with the SR interference phenomenon, it is likely that this disturbance is caused by the electric field originated from lightnings, instead of the Alfvén wave.

3. When the SR disturbance occurs in the ULF band of the electric field, the whistler wave signal is detected in the VLF band. The whistler wave indicates that the electric field energy generated by the lightning and TLEs can cause the disturbance to altitudes as high as the CESE's satellite orbit.

6. To be pedantic, a satellite does not fly; it is in orbit, or it travels in an orbit.

Reply: Thanks for your valuable comment. We revised the statement describing the satellite orbit in the text. We have revised line 109 to say: 'The CSES's orbital position passed near the Luoding station at around 18:33 UT, located on the west side with a horizontal distance of about 200 km. From September to October, around the same orbital positions, i.e., the repeat of the satellite's orbit around Luoding after every five-day cycle, were selected for the background data. The appropriate CSES satellite orbits were found on both September 20 and October 15. No TLEs were captured on October 15. And on September 20, TLEs were only photographed during the 13-15 UT period, with no TLEs appeared when CSES traveled overhead Luoding station at 18:00 UT. The lightnings on September 20 and October 15 were also shown in Figure 2a and 2c. It can be observed that no large-scale thunderstorm area on September 20 and October 15'. We hope the revised description is describing the orbital motion of the CSES's satellite better now.

7. I consider that much more study is required before a paper can be written on this matter.

Reply: Thanks for your valuable comment. In this research, we have found more cases of TLEs affecting the ionospheric electric field. When we chose the satellite orbit within 500 km from the Luoding ground station, there are two cases on May 3 and 23, 2021 that can meet the conditions of this study. A large number of TLEs occurred in the vicinity of Luoding on the two nights, while the satellite orbit was at a horizontal distance of about 200 km from Luoding. The ionospheric Schumann resonance was significantly perturbed in these two cases, and the signal-to-noise ratio decreased with the increase of the electric field energy in a similar way to the present study case. However, we lacked the lightning data near Luoding station at that time, and could only speculate that the parent lightning was about 200 km away from Luoding based on the shooting angle. And unfortunately, some other cases may have no comparison for the same orbit. Due to the limited length of the article, we did not put all the study cases in the main text. We have added the information on more TLEs in the supplementary material (see below), and we hope the reviewer could possibly re-assess our case studies again.

Thanks again for all your comments and questions. We have learned a lot from these valuable suggestions. We have tried our best to express our answers to the critical comments. At the same time, we are mindful that there are a whole lot of works, both observational and theoretical, to be done. We also think that our modest effort to summarize our understanding with the special emphasis on our hard-earned observational measurements is an improvement in the understanding of the subject of Transient Luminous Events. This is why we are still hoping that our manuscript can be published in ACP with the consent of the referee.

**Supplementary materials**

**1. The background SR of local ionosphere:**

[Figure]

Luoding station is located on the northern edge of the thunderstorm's center in South Asia-Southeast Asia region. Thunderstorm activity increases from April to September each year. Ionospheric SR energy enhancement is observed. We selected the background data calculation from April to September. This graph represents the mean of ionospheric SR. From the diagram we can find the SR 1mode and 2mode. We cautiously use it as a background for the SR in 2021. As solar activity intensifies, ionospheric SR appears to be obscured by the strong electric fields (Zhu et al. 2023).

**2. Some additional TLEs cases and parameters studied/examined in this research**

TLEs are difficult to monitor. Some TLEs occur in clusters, and some TLEs occur sporadically at different times. In 2021, a total of 14 cases of TLEs were monitored at the Luoding Station, including 9 cases in May, 1 case in June, 1 case in August, 2 cases in September, and 1 case in October. The CSES satellite orbit travels and passes overhead around this area twice every day at 6:00 UT and 18:00 UT. In addition to the case on 25 September 2021, we obtained 6 cases of TLEs that occurred near Luoding around the time of 18UT. The 6 cases are shown and summarized in the table below. The time and number of photographed TLEs are documented in the table. 'TLE cases' represents the days when these cases appeared in the Luoding station, 'TLE count' represents the total number of TLEs observed around the station, 'Distance' represents the minimum horizontal distance between Luoding station and CSES's satellite orbit, and 'Anomaly range' represents the approximate range of anomalies in satellite orbits.

Finally, we attach specific information on all 6 cases of TLEs.

**TLEs cases and parameters:**

| TLE cases | TLE count | Distance (km) | Anomaly range |
|---|---|---|---|
| May 3, 2021 | 27 | 300 | 13°N-20°N |
| May 10, 2021 | 90 | 1800 | 10°N-25°N |
| May 11, 2021 | 65 | 1300 | 10°N-30°N |
| May 12, 2021 | 26 | 800 | 10°N-16°N |
| May 23, 2021 | 16 | 300 | 17°N-22°N |
| June 5, 2021 | 20 | 1300 | 17°N-22°N |

The details of the TLEs:

(1). TLEs on May 3, 2021

[Figure]

Some TLE cases photographed on May 3, 2021

[Figure]

The PSD of ULF electric field

[Figure]

The SNR of SR 1mode (left) and 2mode (right)

(2). TLEs on May 10, 2021

[Figure]

Some TLE cases photographed on May 10, 2021

[Figure]

The PSD of ULF electric field

[Figure]

The SNR of SR 1mode (left) and 2mode (right)

(3). TLEs on May 11, 2021

[Figure]

Some TLE cases photographed on May 11, 2021

[Figure]

The PSD of ULF electric field

[Figure]

The SNR of SR 1mode (left) and 2mode (right)

(4). TLEs on May 12, 2021

[Figure]

Some TLE cases photographed on May 12, 2021

[Figure]

The PSD of ULF electric field

[Figure]

The SNR of SR 1mode (left) and 2mode (right)

(5). TLEs on May 23, 2021

[Figure]

Some TLE cases photographed on May 23, 2021

[Figure]

The PSD of ULF electric field

[Figure]

The SNR of SR 1mode (left) and 2mode (right)

**(6). TLEs on June 5, 2021**

[Figure]

Some TLE cases photographed on June 5, 2021

[Figure]

The PSD of ULF electric field

[Figure]

The SNR of SR 1mode (left) and 2mode (right)

**Reference of this supplementary materials:**

Zhu, K., Yan, R., Xiong, C., Zheng, L., Zeren, Z., Shen, X., Liu, D., Guan, Y., Liu, C., Xu, S., Lv, F., Guo, F., Zhou, N.: Annual and semi-annual variations of electron density in the topside ionosphere observed by CSES. *Frontiers in Earth Science*, 11, https://doi.org/10.3389/feart.2023.1098483, 2023.

---

## Author Comment (AC2)

We would like to express our cordial thanks to the reviewer for the valuable comments and constructive suggestions. We appreciate the input of the reviewer and hope to make sure that we are incorporating his or her significant inputs to strengthen our analysis. We went through all the comments thoroughly and revised the manuscript accordingly.

This paper as the title describing, presents the first observation of the Transient Luminous Events (TLE) effect on the ionospheric Schumann Resonance (SR), based on the China Seismo-Electromagnetic Satellite. The results are very impressive and novel, which could potentially make significant contribution to the understanding of the electro-coupling between the lower atmosphere and ionosphere. The paper is suitable for ACP and can be published after some revisions and clarifications. The followings are my suggestions of this paper.

1. To further exclude the other possible sources or mechanisms that disturb the power Spectral Density of SR. The author should check if this is the impact of a cosmic gamma flare effect. Also, the authors should consider the possibility of Ionospheric Alfvén Resonator (IAR) illustrated as multi-band Spectral Resonant Structure (SRS) which is associated with tropospheric sprite lightning flashes. Some discussions should be added around this topic to enhance the readability of the paper.

Reply: Thanks for your valuable comment. Regarding other sources or mechanisms of SR Power Spectral Density perturbations, we examined the global SR during the day and determined that the PSD disturbances were regional, short-lived phenomena. In several neighboring orbits, such perturbations occur less frequently. In addition, cosmic rays and solar flares were not significant during the day. After comparing solar activity data such as solar flares and CMEs, it was determined that there was no influence of other extraterrestrial parameters. In order to make the background data more perfect, we excluded the influence of solar flares and cosmic rays, around line 166: 'In order to exclude the influence of other factors, we examined the possible influence of solar activity and high-energy proton events. The data showed that the solar activity was relatively calm during these days, and there were no high-energy particle events.'

For the ionospheric Alfvén resonator (IAR), we compared the Alfvén resonance observed by the Chibis-M satellite, according to 'Electric field signatures of the IAR and Schumann resonance in the upper ionosphere detected by Chibis-M microsatellite' (Dudkin et al. 2014). The Alfvén resonance has a large overlap range with the SR band, and the Alfvén waves may perturb the SR characteristics. Therefore, we compared the SR and Alfvén wave interference phenomena observed in the Eskdalemuir Observatory in the Scottish Borders of the UK (referred 'Observation of Ionospheric Alfvén Resonances at 1–30 Hz and their superposition with the Schumann Resonances' (Beggan et al., 2018)), and conclude that there is some difference between this interference and the PSD perturbation of SR. Both surface and ionospheric Alfvén waves exhibit fingerprints form (Beggan et al. 2018; Dudkin et al., 2014), but CSES observations do not show this feature.

To complete the background of sprites and SRS research, we have added statements around line 62 as: 'The TLEs are closely related to atmospheric and ionospheric electromagnetic activities (Bösinger et al., 2006; Sátori et al., 2013; Shalimov et al., 2011). For example, the spectral structure of the surface SR has been studied by the sprite Q-burst, the electrical impulses generated during TLEs have been analyzed, and the ionospheric Alfvén wave resonances excited by jets have been monitored (Bösinger et al., 2006; Füllekrug et al., 1998; Guha et al., 2017). The Alfvén waves are considered to be controlled by global lightning activity, which will affect the magnetic field in the ionospheric region (Bösinger et al., 2002; Surkov et al., 2013).'

2. If possible, could author provide more cases like the examples shown in the paper which could be appended in the attached files and add a table in the main article. Reply: Thanks for your valuable comment. In this research, we have found more cases of TLEs affecting the ionospheric electric field. When we chose the satellite orbit within 500 km from the Luoding ground station, there are two cases on May 3 and 23, 2021 that can meet the conditions of this study. A large number of TLEs occurred in the vicinity of Luoding on the two nights, while the satellite orbit was at a horizontal distance of about 200 km from Luoding. The ionospheric Schumann resonance was significantly perturbed in these two cases, and the signal-to-noise ratio decreased with the increase of the electric field energy in a similar way to the present study case. However, we lacked the lightning data near Luoding station at that time, and could only speculate that the parent lightning was about 200 km away from Luoding based on the shooting angle. And unfortunately, some other cases may have no comparison for the same orbit. Due to the limited length of the article, we did not put all the study cases in the main text. We have added the information on more TLEs in the supplementary material (see below), and we hope the reviewer could possibly re-assess our case studies again.

3. If possible, the authors may provide more information of the background global space environment and make sure that this is a local phenomenon at a given time and place. Reply: Thanks for your valuable comment. We have statistically analyzed the nighttime electric field data of CSES on 2021. Due to the influence of solar activity, magnetic storms, and surface atmospheric activity, the ionospheric signal in ULF band is greatly perturbed (Balan et al., 2008; Fejer, 1981; Huang et al., 2005; Xiong et al., 2021). In the mid-latitude region of the northern hemisphere in winter, the signal of the Schumann resonance is weak and the frequency is unstable (Ouyang et al., 2015; Sátori et al., 2013; Zhou et al., 2013). This may be due to the concentration of global thunderstorm activity in the southern hemisphere at this time (Hayakawa et al., 2023; Nieckarz et al., 2009). As the center of global thunderstorm activity moves towards the Northern Hemisphere, the Schumann Resonance begins to show enhancement (Hayakawa et al., 2023; Nieckarz et al., 2009). According to our statistical results, there is a higher percentage of cases with more pronounced Schumann resonance (i.e., higher SNR) from May to September. We upload the background image of the PSD for the nighttime ionospheric electric field, i.e., the mean ULF electric field in the Northern Hemisphere for the five months (see the supplementary materials please). The mean ULF appears to have clear first and second modes of SR, representing the ionospheric SR background field.

4. The resolution of the figures can be further enhanced.

Reply: Thanks for your valuable comment. Unfortunately, the sampling frequency of the ULF electric field of CSES is about 0.5Hz, and the sampling distance is about 50Km. Based on the original electric field data, it is difficult to improve the resolution of the image. We appreciate the reviewer's understanding of this shortage of the data.

Some papers maybe related to the topic of this paper which I think should be added.
Maybe missing some latest paper. The authors had better check that.

- Dudkin, V. Pilipenko, V. Korepanov, S. Klimov, R. Holzworth, Electric field signatures of the IAR and Schumann resonance in the upper ionosphere detected by Chibis-M microsatellite, Journal of Atmospheric and Solar-Terrestrial Physics, 10.1016/j.jastp.2014.05.013, 117, (81-87), (2014).
- L. Shalimov, T. Bösinger, Sprite-Producing Lightning-Ionosphere Coupling and Associated Low-Frequency Phenomena, Space Science Reviews, 10.1007/s11214-011-9812-x, 168, 1-4, (517-531), (2011).
- Gabriella Sátori, Michael Rycroft, Pál Bencze, Ferenc Märcz, József Bór, Veronika Barta, Tamás Nagy, Károly Kovács, An Overview of Thunderstorm-Related Research on the Atmospheric Electric Field, Schumann Resonances, Sprites, and the Ionosphere at Sopron, Hungary, Surveys in Geophysics, 10.1007/s10712-013-9222-6, 34, 3, (255-292), (2013).
- Tilmann Bösinger, Ágnes Mika, Sergei L. Shalimov, Christos Haldoupis, Torsten Neubert, Is there a unique signature in the ULF response to sprite-associated lightning flashes?, Journal of Geophysical Research: Space Physics, 10.1029/2006JA011887, 111, A10, (2006).

Reply: Thanks for your valuable comment. We have carefully read these references and corrected the content of the article accordingly. In order to complete the background of the study of ionospheric SR, we have added some references in the introduction and revised parts of the introduction.

We have modified the statements around lines 45-51 as follows:

'This kind of waves with frequencies of 7.8 Hz, 14 Hz, and 20 Hz can resonate

with the phase of the initial wave, that is, the Schumann Resonance (SR) (Balser and Wagner, 1962; Sátori et al., 2013; Schumann, 1952). Because of the existence of SR, some bands of the atmospheric electric field energy will show peaks (e.g., the SR frequency) and valleys (e.g., the non-SR frequency) in the spectral graph (Balser and Wagner, 1962; Galejs, 1970; Sátori et al., 2013; Simões et al., 2011). In 1962, the frequencies of SR were deduced to be  $\omega_1 = 7.8$  Hz,  $\omega_2 = 14.1$  Hz,  $\omega_3 = 20.1$  Hz and  $\omega_4 = 26.6$  Hz for the first time (Balser and Wagner, 1962). With more and more SR monitoring stations around the world, the global SR distribution has been obtained (Ouyang et al., 2015; Sátori et al., 2013; Zhou et al., 2013). In 2011, observations from the Communication and Navigation Outage Forecast System (CNOFS) satellite showed the electric field energy of ionospheric F layer increased in some bands, rather consistent with the surface SR mode (Simões et al., 2011). Further, theoretical calculations suggested that the SR can penetrate the bottom of the ionosphere (Simões et al., 2011; Surkov et al., 2013). Subsequently, the ionospheric SR phenomenon with an energy of  $0.5 \mu V / (m * Hz^{1/2})$  was observed by the Chibis-M satellite, and the results proved the existence of the F-layer SR at low and middle latitudes (Dudkin et al., 2014; Simões et al., 2011; Surkov et al., 2013). These results indicated the presence of SR in the ionosphere could also be provided by the energy from the surface atmosphere.'

We have modified the statements around lines 60-64 as follows:

'The mechanism of sprites and elves involves the electromagnetic field heating the particles at the bottom of the ionosphere, which will thus provide a feedback of huge amount of energy through differential potentials (Boccippio et al., 1995; Mende et al., 1995; Sentman et al., 1995). The TLEs are closely related to atmospheric and ionospheric electromagnetic activities (Bösinger et al., 2006; Sátori et al., 2013; Shalimov et al., 2011). For example, the spectral structure of the surface SR has been studied by the sprite Q-burst, the electrical impulses generated during TLEs have been analyzed, and the ionospheric Alfvén wave resonances excited by jets have been monitored (Bösinger et al., 2006; Füllekrug et al., 1998; Guha et al., 2017). The Alfvén

waves are considered to be controlled by global lightning activity, which will affect the magnetic field in the ionospheric region (Bösinger et al., 2002; Surkov et al., 2013).'

Since the frequency of the Alfvén wave is close to the ULF band, the Alfvén wave generated by the lightning may also cause a perturbation of the electric field. In order to eliminate the Alfvén wave, we have added statement around line 170 as: 'The lightnings will also produce Alfvén waves, which are close to the two modes of SR, and may interfere with the ionospheric SR (Beggan et al., 2018; Dudkin et al., 2014). Therefore, we analyzed the time-frequency characteristics of the interference patterns between Alfvén wave and the Schumann resonance, comparing with the electric field disturbance. The interference is generally manifested as a fingerprint shape. However, the SR anomaly in Fig.3 is quite different from this interference characteristics. Thus, we still believe that this anomaly is not caused by Alfvén wave.'

6. "... CSES are utilized to the study the disturbance of ..." remove "the". Please check every sentence and word to reduce the grammar errors.

Reply: Thanks for your valuable comment. We have corrected the grammatical error at line 184 and checked the whole text. We have modified line 184 to 'In this research, the latest ionospheric electric field data from the CSES are utilized to study the disturbance of ionospheric SR during lightnings and TLEs for the first time.'

**References for this response**

- Balan, N., Alleyne, H., Walker, S., Reme, H., McCrea, I., Aylward. A.: Magnetosphere– ionosphere coupling during the CME events of 07–12 November 2004. *Journal* of Atmospheric and Solar-Terrestrial Physics, 70, 2101-2111, https://doi.org/10.1016/j.jastp.2008.03.015, 2008.
- Balser, M., and Wagner, C. A.: On frequency variations of the Earth-ionosphere cavity modes, *Journal of Geophysical Research*, 67(10), 4081-4083, https://doi.org/10.1029/JZ067i010p04081, 1962.
- Beggan, C. D., Musur, M.: Observation of ionospheric Alfvén resonances at 1–30 Hz and their superposition with the Schumann resonances. *Journal of Geophysical Research: Space Physics*, 123, 4202-4214, https://doi.org/10.1029/2018JA025264, 2018.
- Bösinger, T., Haldoupis, C., Belyaev, P. P., Yakunin, M. N., Semenova, N. V., Demekhov, A. G., and Angelopoulos, V.: Spectral properties of the ionospheric Alfvén resonator observed at a low-latitude station (L= 1.3). *Journal of Geophysical Research: Space Physics*, 107, SIA 4-1-SIA 4-9, https://doi.org/10.1029/2001JA005076, 2002.
- Bösinger, T., Mika, A., Shalimov, S. L., Haldoupis, C., and Neubert, T.: Is there a unique signature in the ULF response to sprite-associated lightning flashes? *Journal of Geophysical Research Space Physics*. 111, A10310, https://doi.org/10.1029/2006JA011887, 2006.
- Dudkin, D., Pilipenko, V., Korepanov, V., Klimov, S., Holzworth, R.: Electric field signatures of the IAR and Schumann resonance in the upper ionosphere detected by Chibis-M microsatellite, *Journal of Atmospheric and Solar-Terrestrial Physics*, 117, 81-87, https://doi.org/10.1016/j.jastp.2014.05.013, 2014.
- Fejer, B. G.: The equatorial ionospheric electric fields. A review. Journal of Atmospheric and terrestrial physics, 43, 377-386, https://doi.org/10.1016/0021-9169(81)90101-X, 1981.
- Füllekrug, M., Fraser-Smith, A. C., and Reising, S. C.: Ultra-slow tails of spriteassociated lightning flashes. *Geophysical Research Letters*, 25, 3497-3500, https://doi.org/10.1029/98GL02590, 1998.
- Galejs, J.: Frequency variations of Schumann resonances, *Journal of Geophysical Research*, 75(16), 3237-3251, https://doi.org/10.1029/JA075i016p03237, 1970.
- Guha, A., Williams, E., Boldi, R., Sátori, G., Nagy, T., Bór, J., Montanya, J., Ortega P.,: Aliasing of the Schumann resonance background signal by sprite-associated Qbursts, *Journal of Atmospheric and Solar-Terrestrial Physics*, 165, 25-37, https://doi.org/10.1016/j.jastp.2017.11.003, 2017.
- Hayakawa, M., Galuk, Y. P., Nickolaenko A. P.: Integrated Schumann Resonance Intensity as an Indicator of the Global Thunderstorm Activity. *Geosciences*, 13(6), 177, https://doi.org/10.3390/geosciences13060177, 2023.
- Huang, C. S., Foster, J. C., Kelley. M. C.: Long-duration penetration of the interplanetary electric field to the low-latitude ionosphere during the main phase of magnetic storms. *Journal of Geophysical Research: Space Physics*, 110(A11), https://doi.org/10.1029/2005JA011202, 2005.

- Nieckarz, Z., Zięba, S., Kułak, A., Michalec A.: Study of the periodicities of lightning activity in three main thunderstorm centers based on Schumann resonance measurements. *Monthly weather review*, 137(12), 4401-4409,https://doi.org/10.1175/2009MWR2920.1, 2009.
- Ouyang, X. Y., Xiao, Z., Hao, Y. Q., Zhang, D. H.: Variability of Schumann resonance parameters observed at low latitude stations in China. *Advance in Space Research*, 56, 1389-1399, http://dx.doi.org/10.1016/j.asr.2015.07.006, 2015.
- Sátori, G., Rycroft, M., Bencze, P., Märcz, F., Bór, J., Barta, V., Nagy, T., and Kovács, K.: An Overview of Thunderstorm-Related Research on the Atmospheric Electric Field, Schumann Resonances, Sprites, and the Ionosphere at Sopron, Hungary, *Surveys in Geophysics*, 34, 255–292, https://doi.org/10.1007/s10712-013-9222-6, 2013.
- Schumann, W. O.: On the free oscillations of a conducting sphere which is surrounded by an air layer and an ionosphere shell (in German), *Zeitschrift für Naturforschung A*, 7(2), 149-154, 1952.
- Shalimov, S. L., Bösinger, T.: Sprite-Producing Lightning-Ionosphere Coupling and Associated Low-Frequency Phenomena, *Space Science Reviews*, 168, 517-531, https://doi.org/10.1007/s11214-011-9812-x, 2011.
- Simões, F., Pfaff, R., and Freudenreich, H.: Satellite observations of Schumann resonances in the Earth's ionosphere, *Geophysical Research Letters*, 38(22), L22101, https://doi.org/10.1029/2011GL049668, 2011.
- Surkov, V. V., Hayakawa, M., Schekotov, A. Y., Fedorov, E. N., and Molchanov, O. A.: Ionospheric Alfvén resonator excitation due to nearby thunderstorms. *Journal* of Geophysical Research: Space Physics, 111, A01303, https://doi.org/10.1029/2005JA011320, 2006.
- Xiong, P., Tong, L., Zhang, K., Shen, X, Battiston, R., Ouzounov, D., Iuppa, R., Crookes, D., Long, C., Zhou H.: Towards advancing the earthquake forecasting by machine learning of satellite data. *Science of The Total Environment*, 771, 145256, https://doi.org/10.1016/j.scitotenv.2021.145256, 2021.
- Zhou, H., Yu, H., Cao B., Qiao, X.: Diurnal and seasonal variations in the Schumann resonance parameters observed at Chinese observatories. *Journal of Atmospheric and Solar-Terrestrial Physics*, 98, 86-96, http://dx.doi.org/10.1016/j.jastp.2013.03.021, 2013.

**Supplementary materials**

**1. The background SR of local ionosphere:**

Luoding station is located on the northern edge of the thunderstorm's center in South Asia-Southeast Asia region. Thunderstorm activity increases from April to September each year. Ionospheric SR energy enhancement is observed. We selected the background data calculation from April to September. This graph represents the mean of ionospheric SR. From the diagram we can find the SR 1mode and 2mode. We cautiously use it as a background for the SR in 2021. As solar activity intensifies, ionospheric SR appears to be obscured by the strong electric fields (Zhu et al. 2023).

**2. Some additional TLEs cases and parameters studied/examined in this research**

TLEs are difficult to monitor. Some TLEs occur in clusters, and some TLEs occur sporadically at different times. In 2021, a total of 14 cases of TLEs were monitored at the Luoding Station, including 9 cases in May, 1 case in June, 1 case in August, 2 cases in September, and 1 case in October. The CSES satellite orbit travels and passes overhead around this area twice every day at 6:00 UT and 18:00 UT. In addition to the case on 25 September 2021, we obtained 6 cases of TLEs that occurred near Luoding around the time of 18UT. The 6 cases are shown and summarized in the table below. The time and number of photographed TLEs are documented in the table. 'TLE cases' represents the days when these cases appeared in the Luoding station, 'Distance' represents the minimum horizontal distance between Luoding station and CSES's satellite orbit, and 'Anomaly range' represents the approximate range of anomalies in satellite orbits.

Finally, we attach specific information on all 6 cases of TLEs.

| TLE cases    | TLE count | Distance (km) | Anomaly range |
|--------------|-----------|---------------|---------------|
| May 3, 2021  | 27        | 300           | 13°N-20°N     |
| May 10, 2021 | 90        | 1800          | 10°N-25°N     |
| May 11, 2021 | 65        | 1300          | 10°N-30°N     |
| May 12, 2021 | 26        | 800           | 10°N-16°N     |
| May 23, 2021 | 16        | 300           | 17°N-22°N     |
| June 5, 2021 | 20        | 1300          | 17°N-22°N     |

**TLEs cases and parameters:**

The details of the TLEs:

(1). TLEs on May 3, 2021

**Some TLE cases photographed on May 3, 2021**

---

## Author Response (AR2)

I recommend the publication of this much improved paper, subject to a few factual corrections.

1. Line 76. Delete "an L shape in the spectrum". Replace with "a falling frequency". Include reference to Helliwell RA, Whistlers and related ionospheric phenomena, 1965, Stanford University Press.

2. Line 160, Fig. 4 caption. Delete "L shape". Replace with "falling frequency".

3. Line 163, 203, 224. Change ULF to ELF. (ELF is generally taken to be between 3 Hz and 3 kHz, with ULF being < 3 Hz.)

4. Fig. 4 caption. Replace whistling and whistle by whistler. Change b to: b. The lightning generated whistler waves power spectral density observed by the magnetometer.

5. Fig. 5. It should be Square root of power that is plotted, not Power, to be consistent with the units shown.

6. Line 148. The peak field strengths of the first and second SR modes are ...

Reply: Thanks for the reviewer's valuable comments and constructive suggestions. We have addressed all the technical corrections in the revised manuscript.

1. We have replaced "L shape" with "falling frequency" at lines 75, 160, and 395.

2. The reference has been added at lines 76 and 323.

3. The ULF has been changed to ELF at lines 25, 89, 163, 178, 203, 205, 214, and 224.

4. The caption of Fig. 4 has been modified to "The lightning whistler waves in obvious falling frequency displayed during orbit NO. 20239, observed on 18:31:36 UT at 16.6°N. a. The lightning whistler waves from the electric field meter. b. The lightning generated whistler waves power spectral density observed by the magnetometer."

5. The unit of Fig. 5 has been corrected.

6. Line 148 has been modified.

Thanks again for all your help in improving our manuscript.